# CONTROLLING GENERATIVE MODELS WITH CONTINUOUS FACTORS OF VARIATIONS

**Antoine Plumerault**[*†]**, Hervé Le Borgne**[*]**, Céline Hudelot**[†]
∗ CEA, LIST, Laboratoire Analyse Sémantique Texte et Image, Gif-sur-Yvette, F-91191 France
† Université Paris-Saclay, CentraleSupélec, 91190, Gif-sur-Yvette, France.

## ABSTRACT

Recent deep generative models are able to provide photo-realistic images as well as visual or textual content embeddings useful to address various tasks of computer vision and natural language processing. Their usefulness is nevertheless often limited by the lack of control over the generative process or the poor understanding of the learned representation. To overcome these major issues, very recent work has shown the interest of studying the semantics of the latent space of generative models. In this paper, we propose to advance on the interpretability of the latent space of generative models by introducing a new method to find meaningful directions in the latent space of any generative model along which we can move to control precisely specific properties of the generated image like the position or scale of the object in the image. Our method does not require human annotations and is particularly well suited for the search of directions encoding simple transformations of the generated image, such as translation, zoom or color variations. We demonstrate the effectiveness of our method qualitatively and quantitatively, both for GANs and variational auto-encoders.

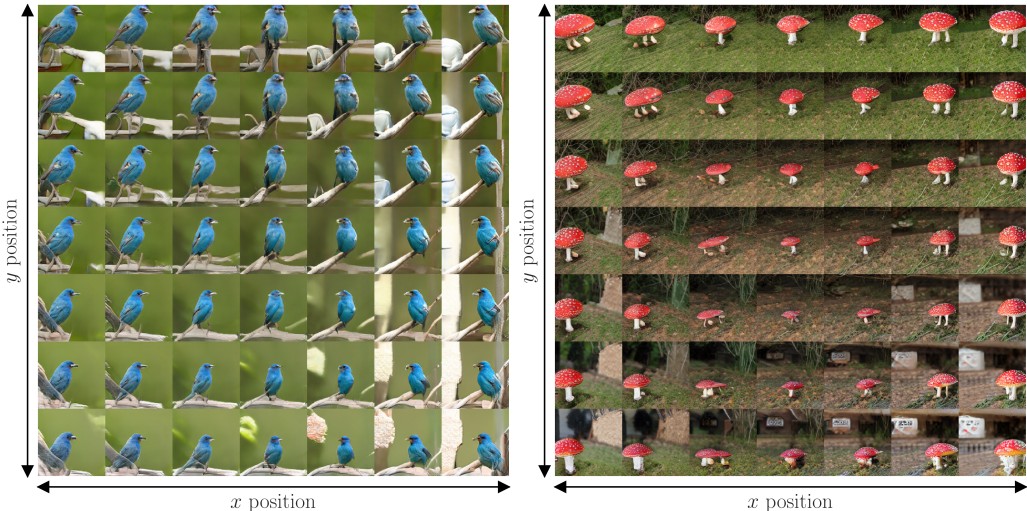

Figure 1: Images generated with our approach and a BigGAN model (Brock et al., 2018), showing that the position of the object can be controlled within the image.

## 1 INTRODUCTION

With the success of recent generative models to produce high-resolution photo-realistic images (Karras et al., 2018; Brock et al., 2018; Razavi et al., 2019), an increasing number of applications are emerging, such as image in-painting, dataset-synthesis, and deep-fakes. However, the use of generative models is often limited by the lack of control over the generated images. More control could be used to

improve existing approaches which aim at generating new training examples (Bowles et al., 2018) by allowing the user to choose more specific properties of the generated images.

First attempts in this direction showed that one can modify an attribute of a generated image by adding a learned vector on its latent code (Radford et al., 2015) or by combining the latent code of two images (Karras et al., 2018). Moreover, the study of the latent space of generative models provides insights about its structure which is of particular interest as generative models are also powerful tools to learn unsupervised data representations. For example, Radford et al. (2015) observed on auto-encoders trained on datasets with labels for some factors of variations, that their latent spaces exhibit a vector space structure where some directions encode the said factors of variations.

We suppose that images result from underlying *factors of variation* such as the presence of objects, their relative positions or the lighting of the scene. We distinguish two categories of factors of variations. *Modal factors of variation* are discrete values that correspond to isolated clusters in the data distribution, such as the category of the generated object. On the other hand, the size of an object or its position are described by *Continuous factors of variations*, expressed in a range of possible values. As humans, we naturally describe images by using *factors of variations* suggesting that they are an efficient representation of natural images. For example, to describe a scene, one likely enumerates the objects seen, their relative positions and relations and their characteristics (Berg et al., 2012). This way of characterizing images is also described in Krishna et al. (2016). Thus, explaining the latent space of generative models through the lens of factors of variation is promising. However, the control over the image generation is often limited to discrete factors and requires both labels and an encoder model. Moreover, for continuous factors of variations described by a real parameter $t$, previous works do not provide a way to get precise control over $t$.

In this paper, we propose a method to find meaningful directions in the latent space of generative models that can be used to control precisely specific continuous factors of variations while the literature has mainly tackled semantic labeled attributes like gender, emotion or object category (Radford et al., 2015; Odena et al., 2016). We test our method on image generative models for three factors of variation of an object in an image: vertical position, horizontal position and scale. Our method has the advantage of not requiring a labeled dataset nor a model with an encoder. It could be adapted to other factors of variations such as rotations, change of brightness, contrast, color or more sophisticated transformations like local deformations. However, we focused on the position and scale as these are quantities that can be evaluated, allowing us to measure quantitatively the effectiveness of our method. We demonstrate both qualitatively and quantitatively that such directions can be used to control precisely the generative process and show that our method can reveal interesting insights about the structure of the latent space. Our main contributions are:

- We propose a method to find interpretable directions in the latent space of generative models, corresponding to parametrizable continuous *factors of variations* of the generated image.
- We show that properties of generated images can be controlled precisely by sampling latent representations along linear directions.
- We propose a novel reconstruction loss for inverting generative models with gradient descent.
- We give insights of why inverting generative models with optimization can be difficult by reasoning about the geometry of the natural image manifold.
- We study the impacts of disentanglement on the ability to control the generative models.

## 2 LATENT SPACE DIRECTIONS OF A FACTOR OF VARIATION

We argue that it is easier to modify a property of an image than to obtain a label describing that property. For example, it is easier to translate an image than to determine the position of an object within said image. Hence, if we can determine the latent code of a transformed image, we can compute its difference with the latent code of the original image to find the direction in the latent space which corresponds to this specific transformation as in Radford et al. (2015).

Let us consider a generative model $G : z \in \mathcal{Z} \to \mathcal{I}$, with $\mathcal{Z}$ its latent space of dimension $d$ and $\mathcal{I}$ the space of images, and a transformations $\mathcal{T}_t : \mathcal{I} \to \mathcal{I}$ characterized by a continuous parameter $t$. For example if $\mathcal{T}$ is a rotation, then $t$ could be the angle, and if $\mathcal{T}$ is a translation, then $t$ could be a component of the vector of the translation in an arbitrary frame of reference. Let $z_0$ be a vector

of $\mathcal{Z}$ and $I = G(z_0)$ a generated image. Given a transformation $\mathcal{T}_T$, we aim at finding $z_T$ such that $G(z_T) \approx \mathcal{T}_T(I)$ to then use the difference between $z_0$ and $z_T$ in order to estimate the direction encoding the factor of variation described by $\mathcal{T}$.

## 2.1 LATENT SPACE TRAJECTORIES OF AN IMAGE TRANSFORMATION

Given an image $I \in \mathcal{I}$, we want to determine its latent code. When no encoder is available we can search an approximate latent code $\hat{z}$ that minimizes a reconstruction error $\mathcal{L}$ between $I$ and $\hat{I} = G(\hat{z})$ ($\hat{I}$ can be seen as the projection of $I$ on $G(\mathcal{Z})$) i.e.

$$\hat{z} = \arg\min_{z \in \mathcal{Z}} \mathcal{L}(I, G(z)) \tag{1}$$

Solving this problem by optimization leads to solutions located in regions of low likelihood of the distribution used during training. It causes the reconstructed image $\hat{I} = G(\hat{z})$ to look unrealistic[1]. Since $z$ follows a normal distribution $\mathcal{N}(0, I_d)$ in a $d$-dimensional space, we have $||z|| \sim \chi_d$. Thus, $\lim_{d \to +\infty} \mathbb{E}[||z||] = \sqrt{d}$ and $\lim_{d \to +\infty} \text{Var}(||z||) = 0$. Hence, when $d$ is large, the norm of $z$ is approximately equal to $\sqrt{d}$. This can be used to regularize the optimization by constraining $z$ to verify $||z|| \leq \sqrt{d}$:

$$\hat{z} = \arg\min_{z \in \mathcal{Z}, ||z|| \leq \sqrt{d}} \mathcal{L}(I, G(z)) \tag{2}$$

### 2.1.1 CHOICE OF THE RECONSTRUCTION ERROR $\mathcal{L}$

One of the important choice regarding this optimization problem is that of $\mathcal{L}$. In the literature, the most commonly used are the pixel-wise Mean Squared Error (MSE) and the pixel-wise cross-entropy as in Lipton & Tripathi (2017) and Creswell & Bharath (2016). However in practice, pixel-wise losses are known to produce blurry images. To address this issue, other works have proposed alternative reconstruction errors. However, they are based on an alternative neural network (Boesen Lindbo Larsen et al., 2015; Johnson et al., 2016) making them computationally expensive.

The explanation usually given for the poor performance of pixel-wise mean square error is that it favors the solution which is the expected value of all the possibilities (Mathieu et al., 2015)[2]. We propose to go deeper into this explanation by studying the effect of the MSE on images in the frequency domain. In particular, our hypothesis is that due to its limited capacity and the low dimension of its latent space, the generator can not produce arbitrary texture patterns as the manifold of textures is very high dimensional. This uncertainty over texture configurations explains why textures are reconstructed as uniform regions when using pixel-wise errors. In Appendix A, by expressing the MSE in the Fourier domain and assuming that the phase of high frequencies cannot be encoded in the latent space, we show that the contribution of high frequencies in such a loss is proportional to their square magnitude pushing the optimization to solutions with less high frequencies, that is to say more blurry. In order to get sharper results we therefore propose to reduce the weight of high frequencies into the penalization of errors with the following loss:

$$\mathcal{L}(I_1, I_2) = ||\mathcal{F}\{I_1 - I_2\}\mathcal{F}\{\sigma\}||^2 = ||(I_1 - I_2) * \sigma||^2 \tag{3}$$

where $\mathcal{F}$ is the Fourier transform, $*$ is the convolution operator and $\sigma$ is a Gaussian kernel. With a reduced importance given to the high frequencies to determine $\hat{z}$ when one uses this loss in equation 2, it allows to benefit from a larger range of possibilities for $G(z)$, including images with more details (i.e with more high frequencies) and appropriate texture to get more realistic generated images. A qualitative comparison to some reconstruction errors and choices of $\sigma$ can be found in Appendix C. We also report a quantitative comparison to other losses, based on the Learned Perceptual Image Patch Similarity (LPIPS), proposed by Zhang et al. (2018).

---

[1] We could have used a $L^2$ penalty on the norm of $z$ to encode a centered Gaussian prior on the distribution of $z$. However the $L^2$ penalty requires an additional hyper-parameter $\beta$ that can be difficult to choose.

[2] Indeed, if we model the value of pixel by a random variable x then $\arg\min_x \mathbb{E}[(x - \mathrm{x})^2] = \mathbb{E}[\mathrm{x}]$. In fact, this problem can easily generalized at every pixel-wise loss if we assume that nearby pixels follows approximately the same distribution as $\arg\min_x \mathbb{E}[\mathcal{L}(x, \mathrm{x})]$ will have the same value for nearby pixels.

**Algorithm 1:** Create a dataset of trajectories in the latent space which corresponds to a transformation $\mathcal{T}$ in the pixel space. The transformation is parametrized by a parameter $\delta t$ which controls a degree of transformation. We typically use $N = 10$ with $(\delta t_n)_{(0 \leq n \leq N)}$ distributed regularly on the interval $[0, T]$. Note that $z_0$ and $\delta t_n$ are retained in $D$ at each step to train the model of Section 2.2.

---

**Input:** number of trajectories $S$, generator $G$, transformation function $\mathcal{T}$, trajectories length $N$, threshold $\Theta$.

**Result:** dataset of trajectories $D$

$D \leftarrow \{\}$ ;

**for** $i \in [\![1, S]\!]$ **do**

    $z_0 \sim \mathcal{N}(0, I)$ ;

    $I_0 \leftarrow G(z_0)$ ;

    $z_{\delta t} \leftarrow z_0$ ;

    **for** $n \in [1, N]$ **do**

        $z_{\delta t} \leftarrow \arg\min_z \mathcal{L}(G(z), \mathcal{T}_{\delta t_n}(I_0))$ ;

        **if** $\mathcal{L}(G(z), \mathcal{T}_{\delta t_n}(I_0)) < \Theta$ **then**

            $D \leftarrow D \cup \{(z_0, z_{\delta t}, \delta t_n)\}$ ;

        **end**

    **end**

**end**

---

### 2.1.2 RECURSIVE ESTIMATION OF THE TRAJECTORY

Using equation 2, our problem of finding $z_T$ such that $G(z_T) \approx \mathcal{T}_T(I)$, given transformation $\mathcal{T}_T$, can be solve through the following optimization problem:

$$z_T = \arg\min_{z \in \mathcal{Z}, ||z|| \leq \sqrt{d}} \mathcal{L}(G(z), \mathcal{T}_T(I)) \tag{4}$$

In practice, this problem is difficult and an "unlucky" initialization can lead to a very slow convergence. Zhu et al. (2016) proposed to use an auxiliary network to estimate $z_T$ and use it as initialization. Training a specific network to initialize this problem is nevertheless costly. One can easily observe that a linear combination of natural images is usually not a natural image itself, this fact highlights the highly curved nature of the manifold of natural images in pixel space. In practice, the trajectories corresponding to most transforms in pixel space may imply small gradients of the loss that slowdown the convergence of problem of Eq. ( 2) (see Appendix D). To address this, we guide the optimization on the manifold by decomposing the transformation $\mathcal{T}_T$ into smaller transformations $[\mathcal{T}_{\delta t_0}, \ldots, \mathcal{T}_{\delta t_N}]$ such that $\mathcal{T}_{\delta t_0 = 0} = Id$ and $\delta t_N = T$ and solve sequentially:

$$z_n = \arg\min_{z \in \mathcal{Z}} \mathcal{L}\left(G\left(z; z_{init} = z_{n-1}\right), \mathcal{T}_{\delta t_n}\left(G\left(z_0\right)\right)\right) \quad \text{for } n = 1, \ldots, N \tag{5}$$

each time initializing $z$ with the result of the previous optimization. In comparison to Zhu et al. (2016), our approach does not require extra training and can thus be used directly without training a new model. We compare qualitatively our method to a naive optimization in Appendix C.

A transformation on an image usually leads to undefined regions in the new image (for instance, for a translation to the right, the left hand side is undefined). Hence, we ignore the value of the undefined regions of the image to compute $\mathcal{L}$. Another difficulty is that often the generative model cannot produce arbitrary images. For example a generative model trained on a given dataset is not expected to be able to produce images where the object shape position is outside of the distribution of object shape positions in the dataset. This is an issue when applying our method because as we generate images from a random start point, we have no guarantee that the transformed images is still on the data manifold. To reduce the impact of such outliers, we discard latent codes that give a reconstruction error above a threshold in the generated trajectories. In practice, we remove one tenth of the latent codes which leads to the worst reconstruction errors. It finally results into Algorithm 1 to generate trajectories in the latent space.

## 2.2 ENCODING MODEL OF THE FACTOR OF VARIATION IN THE LATENT SPACE.

After generating trajectories with Algorithm 1, we need to define a model which describes how factors of variations are encoded in the latent space. We make the core hypothesis that the parameter $t$ of a specific factor of variations can be predicted from the coordinate of the latent code along an axis $\boldsymbol{u}$, thus we pose a model $f : \mathcal{Z} \to \mathbb{R}$ of the form $t = f(\boldsymbol{z}) = g(\langle \boldsymbol{z}, \boldsymbol{u} \rangle)$, with $g : \mathbb{R} \to \mathbb{R}$ and $\langle \cdot, \cdot \rangle$ the euclidean scalar product in $\mathbb{R}^d$.

When $g$ is a monotonic differentiable function, we can without loss of generality, suppose that $\|u\| = 1$ and that $g$ is an increasing function. Under these conditions, the distribution of $t = g(\langle \boldsymbol{z}, \boldsymbol{u} \rangle)$ when $\boldsymbol{z} \sim \mathcal{N}(0, \boldsymbol{I})$ is given by $\varphi : \mathbb{R} \to \mathbb{R}_+$:

$$\varphi(t) = \mathcal{N}(g^{-1}(t); 0, 1) \frac{d}{dt} g^{-1}(t) \tag{6}$$

For example, consider the dSprite dataset (Matthey et al., 2017) and the factor corresponding to the horizontal position of an object $x$ in an image, we have $x$ that follows a uniform distribution $\mathcal{U}([-0.5, 0.5])$ in the dataset while the projection of $\boldsymbol{z}$ onto an axis $\boldsymbol{u}$ follows a normal distribution $\mathcal{N}(0, 1)$. Thus, it is natural to adopt $g : \mathbb{R} \to [-0.5, 0.5]$ and for $x = g(\langle \boldsymbol{z}, \boldsymbol{u} \rangle)$:

$$
\begin{aligned}
\varphi(x) &= \mathcal{U}(x, [-0.5, 0.5]) = \mathcal{N}(g^{-1}(x); 0, 1) \frac{d}{dx} g^{-1}(x) &&\Longleftrightarrow \\
1 &= \mathcal{N}(\langle \boldsymbol{z}, \boldsymbol{u} \rangle; 0, 1) \frac{d}{dx} g^{-1}(g(\langle \boldsymbol{z}, \boldsymbol{u} \rangle)) &&\Longleftrightarrow \\
\frac{1}{\frac{d}{dx} g^{-1}(g(\langle \boldsymbol{z}, \boldsymbol{u} \rangle))} &= \frac{d}{dx} g(\langle \boldsymbol{z}, \boldsymbol{u} \rangle) = \mathcal{N}(\langle \boldsymbol{z}, \boldsymbol{u} \rangle; 0, 1) &&\Longleftrightarrow \\
g(\langle \boldsymbol{z}, \boldsymbol{u} \rangle) &= \frac{1}{2} \operatorname{erf}\left(\frac{\langle \boldsymbol{z}, \boldsymbol{u} \rangle}{\sqrt{2}}\right)
\end{aligned}
\tag{7}
$$

However, in general, the distribution of the parameter $t$ is not known. One can adopt a more general parametrized model $g_\theta$ of the form:

$$t = f_{(\theta, \boldsymbol{u})}(\boldsymbol{z}) = g_\theta(\langle \boldsymbol{u}, \boldsymbol{z} \rangle) \quad \text{with} \quad ||\boldsymbol{u}|| = 1 \tag{8}$$

with $g_\theta : \mathbb{R} \to \mathbb{R}$ and $(\theta, \boldsymbol{u})$ trainable parameters of the model. We typically used piece-wise linear functions for $g_\theta$.

However, this model cannot be trained directly as we do not have access to $t$ (in the case of horizontal translation the $x$-coordinate for example) but only to the difference $\delta t = t_{G(\boldsymbol{z}_{\delta t})} - t_{G(\boldsymbol{z}_0)}$ between an image $G(\boldsymbol{z}_0)$ and its transformation $G(\boldsymbol{z}_{\delta t})$ ($\delta x$ or $\delta y$ in the case of translation). We solve this issue by modeling $\delta t$ instead of $t$:

$$\delta t = f_{(\theta, \boldsymbol{u})}(\boldsymbol{z}_{\delta t}) - f_{(\theta, \boldsymbol{u})}(\boldsymbol{z}_0) \quad \text{with} \quad ||\boldsymbol{u}|| = 1 \quad \text{and} \quad g_\theta(0) = 0 \tag{9}$$

Hence, $\boldsymbol{u}$ and $\theta$ are estimated by training $f_{(\theta, \boldsymbol{u})}$ to minimize the MSE between $\delta_t$ and $f_{(\theta, \boldsymbol{u})}(\boldsymbol{z}_{\delta t}) - f_{(\theta, \boldsymbol{u})}(\boldsymbol{z}_0)$ with gradient descent on a dataset produced by Algorithm 1 for a given transformation.

An interesting application of this method is the estimation of the distribution of the images generated by $G$ by using Equation 6. With the knowledge of $g_\theta$ we can also choose how to sample images. For instance, let say that we want to have $t \sim \phi(t)$, with $\phi : \mathbb{R} \to \mathbb{R}_+$ an arbitrary distribution, we can simply transform $z \sim \mathcal{N}(0, 1)$ as follows:

$$\boldsymbol{z} \leftarrow \boldsymbol{z} - \langle \boldsymbol{z}, \boldsymbol{u} \rangle \boldsymbol{u} + (h_\phi \circ \psi)(\langle \boldsymbol{z}, \boldsymbol{u} \rangle) \boldsymbol{u} \tag{10}$$

with $h_\phi : [0, 1] \to \mathbb{R}$ and $\psi$ such that:

$$\psi(x) = \int_{-\infty}^{x} \mathcal{N}(t; 0, 1) dt \;\; ; \;\; h_\phi^{-1}(x) = \int_{-\infty}^{x} \phi(g_\theta(t)) \frac{d}{dt} g_\theta(t) dt \tag{11}$$

These results are interesting to bring control not only on a single output of a generative model but also on the distribution of its outputs. Moreover, since generative models reflect the datasets on which they have been trained, the knowledge of these distributions could be applied to the training dataset to reveal potential bias.

## 3 EXPERIMENTS

**Datasets:** We performed experiments on two datasets. The first one is dSprites (Matthey et al., 2017), composed of 737280 binary $64 \times 64$ images containing a white shape on a dark background. Shapes can vary in position, scale and orientations making it ideal to study disentanglement. The second dataset is ILSVRC (Russakovsky et al., 2015), containing $1.2M$ natural images from one thousand different categories.

**Implementation details:** All our experiments have been implemented with TensorFlow 2.0 (Abadi et al., 2015) and the corresponding code is available on github `here`. We used a BigGAN model (Brock et al., 2018) whose weights are taken from TensorFlow-Hub allowing easy reproduction of our results. The BigGAN model takes two vectors as inputs: a latent vector $z \in \mathbb{R}^{128}$ and a one-hot vector to condition the model to generate images from one category. The latent vector $z$ is then split into six parts which are the inputs at different scale levels in the generator. The first part is injected at the bottom layer while next parts are used to modify the *style* of the generated image thanks to Conditional Batch Normalization layers (de Vries et al., 2017). We also trained several $\beta$-VAEs (Higgins et al., 2017) to study the importance of disentanglement in the process of controlling generation. The exact $\beta$-VAE architecture used is given in Appendix B. The models were trained on dSprites (Matthey et al., 2017) with an Adam optimizer during $1e5$ steps with a batch size of 128 images and a learning rate of $5e-4$.

### 3.1 QUANTITATIVE EVALUATION METHOD

Evaluating quantitatively the effectiveness of our method on complex datasets is intrinsically difficult as it is not always trivial to measure a factor of variation directly. We focused our analysis on two factors of variations: position and scale. On simple datasets such as dSprites, the position of the object can be estimated effectively by computing the barycenter of white pixels. However, for natural images sampled with the BigGAN model, we have to use first saliency detection on the generated image to produce a binary image from which we can extract the barycenter. For saliency detection, we used the model provided by Hou et al. (2016) which is implemented in the PyTorch framework (Paszke et al., 2017). The scale is evaluated by the proportion of salient pixels. The evaluation procedure is:

1. Get the direction $u$ which should describe the chosen factor of variation with our method.
2. Sample latent codes $z$ from a standard normal distribution.
3. Generate images with latent code $z - \langle z, u \rangle u + tu$ with $t \in [-T, T]$.
4. Estimate the real value of the factor of variation for all the generated images.
5. Measure the standard deviation of this value with respect to $t$.

Jahanian et al. (2019) proposed an alternative method for quantitative evaluation that relies on an object detector. Similarly to us, it allows an evaluation for $x$ and $y$ shift as well as scale but is restricted to image categories that can be recognized by a detector trained on some categories of ILSVRC. The proposed approach is thus more generic.

### 3.2 RESULTS ON BIGGAN

We performed quantitative analysis on ten chosen categories of objects of ILSVRC, avoiding non actual objects such as "beach" or 'cliff". Results are presented in Figure 2 (top). We observe that for the chosen categories of ILSVRC, we can control the position and scale of the object relatively precisely by moving along directions of the latent space found by our method.

However, one can still wonder whether the directions found are independent of the category of interest. To answer this question, we merged all the datasets of trajectories into one and learned a common direction on the resulting datasets. Results for the ten test categories are shown in Figure 2 (bottom). This figure shows that the directions which correspond to some factors of variations are indeed shared between all the categories. Qualitative results are also presented in Figure 3 for illustrative purposes. We also checked which parts of the latent code are used to encode position and scale. Indeed, BigGAN uses hierarchical latent code which means that the latent code is split into six parts

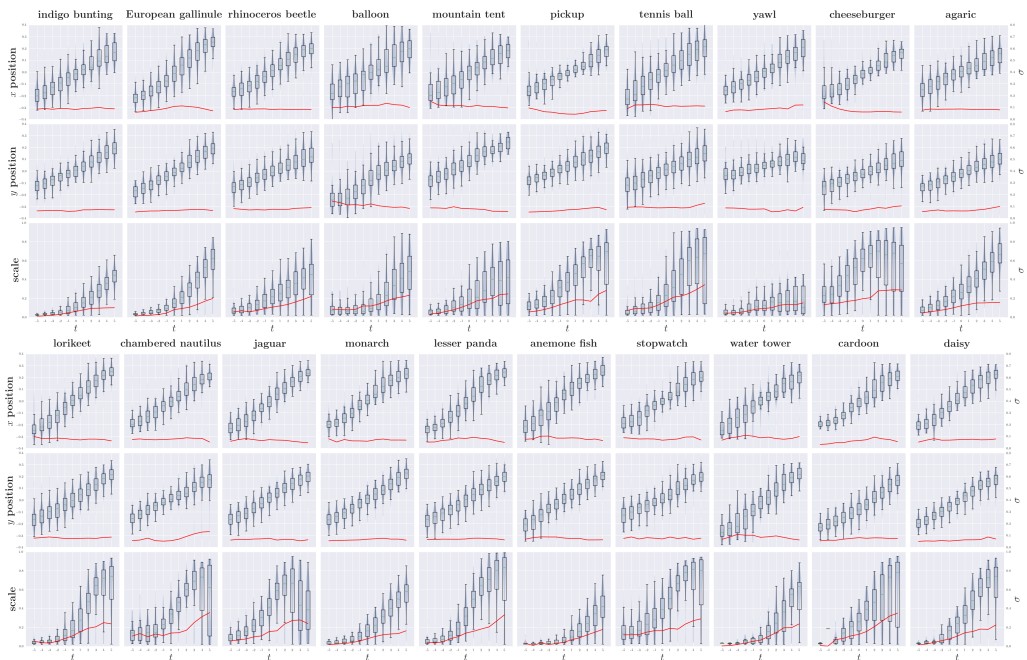

Figure 2: Quantitative results on the ten categories of the ILSVRC dataset used for training (Top) and for ten other categories used for validation (Bottom) for three geometric transformations: horizontal and vertical translations and scaling. In blue, the distribution of the measured transformation parameter and in red the standard deviation of the distribution with respect to $t$. Note that for large scales the algorithm seems to fail. However, this phenomenon is very likely due to the poor performance of the saliency model when the object of interest covers almost the entire image (scale $\approx 1.0$). (best seen with zoom)

which are injected at different level of the generator. We wanted to see by which part of the latent code these directions are encoded. The squared norm of each part of the latent code is reported in Figure 4 for horizontal position, vertical position and scale. This figure shows that the directions corresponding to *spatial factors of variations* are mainly encoded in the first part of the latent code. However, for the $y$ position, the contribution of level 5 is higher than for the $x$ position and the scale. We suspect that it is due to correlations between the vertical position of the object in the image and its background that we introduced by transforming the objects because the background is not invariant by vertical translation because of the horizon.

### 3.3 THE IMPORTANCE OF DISENTANGLED REPRESENTATIONS

To test the effect of disentanglement on the performance of our method, we trained several $\beta$-VAE (Higgins et al., 2017) on dSprites (Matthey et al., 2017), with different $\beta$ values. Indeed, $\beta$-VAE are known for having more disentangled latent spaces as the regularization parameter $\beta$ increases. Results can be seen in Figure 5. The figure shows that it is possible to control the position of the object on the image by moving in the latent space along the direction found with our method. As expected, the effectiveness of the method depends on the degree of disentanglement of the latent space since the results are better with a larger $\beta$. Indeed we can see on Figure 5 that as $\beta$ increases, the standard deviation decreases (red curve), allowing a more precise control of the position of the generated images. This observation motivates further the interest of disentangled representations for control on the generative process.

## 4 RELATED WORKS

Our work aims at finding interpretable directions in the latent space of generative models to control their generative process. We distinguish two families of generative models: GAN-like models which

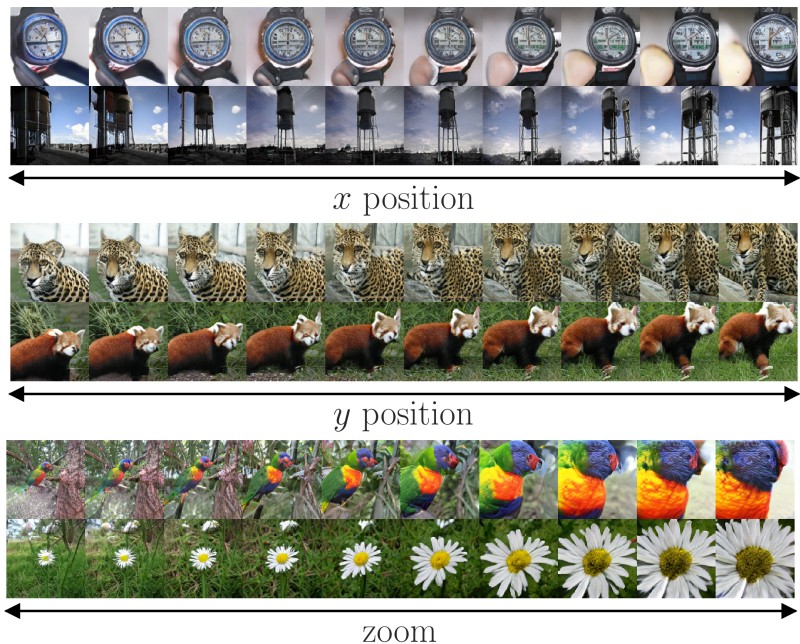

Figure 3: Qualitative results for some categories of ILSVRC dataset for three geometric transformations: horizontal and vertical translations and scaling.

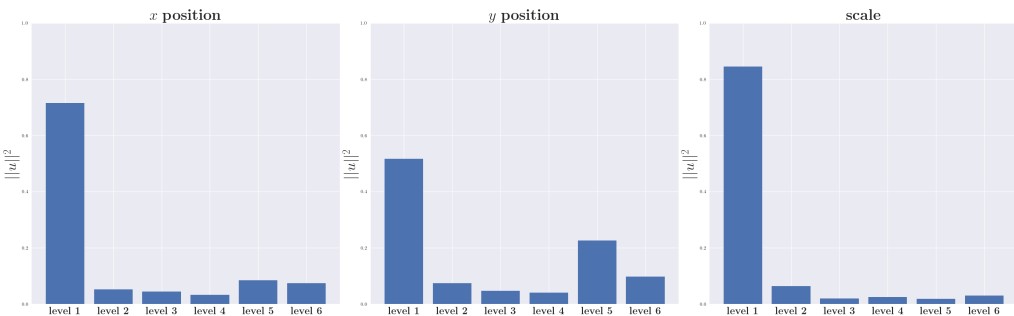

Figure 4: Squared norm of each part of the latent code for horizontal position, vertical position and scale.

do not provide an explicit way to get the latent representation of an image and auto-encoders which provide an encoder to get the latent representation of images. From an architectural point of view, conditional GANs (Odena et al., 2016) allows the user to choose the category of a generated object or some chosen properties of the generated image but this approach requires a labeled dataset and use a model which is explicitly designed to allow this control. Similarly regarding VAE, Engel et al. (2018) identified that they suffer from a trade-off between reconstruction accuracy and sample plausibility and proposed to identify regions of the latent space that correspond to plausible samples to improve reconstruction accuracy. They also use conditional reconstruction to control the generative process. In comparison to these approaches, our method does not directly requires labels. With InfoGan, Chen et al. (2016) shows that adding a code to the the input of the GAN generator and optimizing with an appropriate regularization term leads to disentangle the latent space and make possible to find a posteriori meaningfully directions. In contrast, we show that it is possible to find such directions in several generative models, without changing the learning process (our approach could even be applied to InfoGAN) and with an a priori knowledge of the factor of variation sought. More recently, Bau et al. (2018) analyze the activations of the network's neurons to determine those that result in the presence of an object in the generated image, and thus allows to control such a presence. In contrast, our work focuses on the latent space and not on the intermediate activations inside the generator.

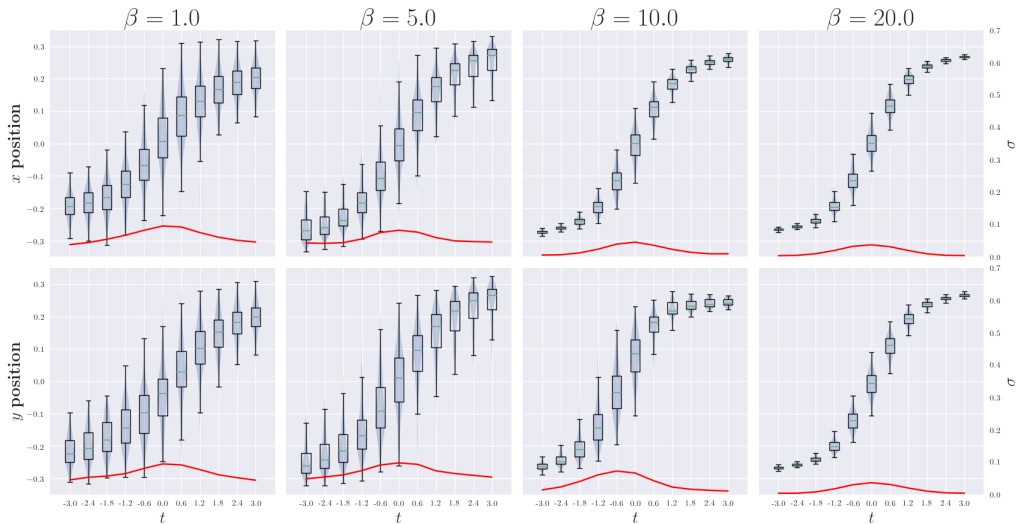

Figure 5: Results of our evaluation procedure with four $\beta$-VAE for $\beta = 1, 5, 10, 20$. Note the erf shape of the results which indicates that the distribution of the shape positions has been correctly learned by the VAE. See Figure 2 for additional information on how to read this figure.

One of our contribution and a part of our global method is a procedure to find the latent representation of an image when an encoder is not available. Several previous works have studied how to invert the generator of a GAN to find the latent code of an image. Creswell & Bharath (2016) showed on simple datasets (MNIST (Lecun et al., 1998) and Omniglot (Lake et al., 2015)) that this inversion process can be achieved by optimizing the latent code to minimize the reconstruction error between the generated image and the target image. Lipton & Tripathi (2017) introduced tricks to improve the results on a more challenging dataset (CelebA (Liu et al., 2015)). However we observed that these methods fail when applied on a more complex datasets (ILSVRC (Russakovsky et al., 2015)). The reconstruction loss introduced in Section 2.1.1 is adapted to this particular problem and improves the quality of reconstructions significantly. We also theoretically justify the difficulties to invert a generative model, compared to other optimization problems. In the context of vector space arithmetic in a latent space, White (2016) argues that replacing a linear interpolation by a spherical one allows to reduce the blurriness as well. This work also propose an algorithmic data augmentation, named "synthetic attribute", to generate image with less noticeable blur with a VAE. In contrast, we act directly on the loss.

The closest works were released on ArXiv very recently (Goetschalckx et al., 2019; Jahanian et al., 2019) indicating that finding interpretable directions in the latent space of generative models to control their output is of high interest for the community. In these papers, the authors describe a method to find interpretable directions in the latent space of the BigGAN model (Brock et al., 2018). If their method exhibits similarities with ours (use of transformation, linear trajectories in the latent space), it also differs on several points. From a technical point of view our training procedure differs in the sense that we first generate a dataset of interesting trajectories to then train our model while they train their model directly. Our evaluation procedure is also more general as we use a saliency model instead of a MobileNet-SSD v1 Liu et al. (2016) trained on specific categories of the ILSVRC dataset allowing us to measure performance on more categories. We provide additional insight on how auto-encoders can also be controlled with the method, the impact of disentangled representations on the control and on the structure of the latent space of BigGAN. Moreover we also propose an alternative reconstruction error to invert generators. However, the main difference we identify between the two works is the model of the latent space used. Our model allows a more precise control over the generative process and can be being adapted to more cases.

## 5    CONCLUSIONS

Generative models are increasingly more powerful but suffer from little control over the generative process and the lack of interpretability in their latent representations. In this context, we propose a method to extract meaningful directions in the latent space of such models and use them to control precisely some properties of the generated images. We show that a linear subspace of the latent space of BigGAN can be interpreted in term of intuitive factors of variation (namely translation and scale). It is an important step toward the understanding of the representations learned by generative models.

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

## A PENALTY ON THE AMPLITUDE OF FREQUENCIES DUE TO MSE

In Section 2.1, we consider a target image $I \in \mathcal{I}$ and a generated image $\hat{I} = G(\hat{z})$ to be determined according to a reconstruction loss $\mathcal{L}$ (Equation 1). Let us note $\mathcal{F}\{\cdot\}$ the Fourier transform. If $\mathcal{L}$ is the usual MSE, from the Plancherel theorem, we have $||\hat{I} - I||^2 = ||\mathcal{F}\{\hat{I}\} - \mathcal{F}\{I\}||^2$. Let us consider a particular frequency $\omega$ in the Fourier space and compute its contribution to the loss. The Fourier transform of $I$ (resp. $\hat{I}$) having a magnitude $r$ (resp. $\hat{r}$) and a phase $\theta$ (resp. $\hat{\theta}$) at $\omega$, we have:

$$
\begin{aligned}
|\mathcal{F}\{\hat{I}\}(\omega) - \mathcal{F}\{I\}(\omega)|^2 &= |\hat{r}e^{i\hat{\theta}} - re^{i\theta}|^2 \\
&= (\hat{r}cos(\hat{\theta}) - rcos(\theta))^2 + (\hat{r}sin(\hat{\theta}) - rsin(\theta))^2 \\
&= \hat{r}^2 + r^2 - 2\hat{r}r\left(cos(\hat{\theta})cos(\theta) + sin(\hat{\theta})sin(\theta)\right) \\
&= \hat{r}^2 + r^2 - 2\hat{r}r\left(cos(\hat{\theta})cos(\theta) + sin(\hat{\theta})sin(\theta)\right)
\end{aligned} \tag{12}
$$

If we model the disability of the generator to model every high frequency patterns as an uncertainty on the phase of high frequency of the generated image, i.e by posing $\hat{\theta} \sim \mathcal{U}([0, 2\pi])$, the expected value of the high frequency contributions to the loss is equal to:

$$
\mathbb{E}\left[|\mathcal{F}\{\hat{I}\}(\omega) - \mathcal{F}\{I\}(\omega)|^2\right] = \hat{r}^2 + r^2 - 2\hat{r}r\left(\underbrace{\mathbb{E}\left[cos(\hat{\theta})\right]}_{=0}cos(\theta) + \underbrace{\mathbb{E}\left[sin(\hat{\theta})\right]}_{=0}sin(\theta)\right) \tag{13}
$$

$$
= \hat{r}^2 + r^2
$$

The term $r^2$ is a constant w.r.t the optimization of $\mathcal{L}$ and can thus be ignored. The contribution to the total loss $\mathcal{L}$ thus directly depends on $\hat{r}^2$. While minimizing $\mathcal{L}$, the optimization process tends to favour images $\hat{I} = G(\hat{z})$ with smaller magnitudes in the high frequencies, that is to say smoother images, with less high frequencies.

## B $\beta$-VAE ARCHITECTURE

The $\beta$-VAE framework was introduced by Higgins et al. (2017) to discover interpretable factorised latent representations for images without supervision. For our experiments, we designed a simple convolutional VAE architecture to generate images of size 64x64, the decoder network is the opposite of the encoder with transposed convolutions.

| Encoder |
|---|
| **Convolution + ReLU** filters=32 size=4 stride=2 pad=SAME |
| **Convolution + ReLU** filters=32 size=4 stride=2 pad=SAME |
| **Convolution + ReLU** filters=32 size=4 stride=2 pad=SAME |
| **Convolution + ReLU** filters=32 size=4 stride=2 pad=SAME |
| **Dense + ReLU** units=256 |
| **Dense + ReLU** units=256 |
| $\mu$: **Dense + Identity** $\sigma$: **Dense + Exponential** units=10 |

| Decoder |
|---|
| **Dense + ReLU** units=256 |
| **Dense + ReLU** units=256 |
| **Reshape** shape=4x4x32 |
| **Transposed Convolution + ReLU** filters=32 size=4 stride=2 pad=SAME |
| **Transposed Convolution + ReLU** filters=32 size=4 stride=2 pad=SAME |
| **Transposed Convolution + ReLU** filters=32 size=4 stride=2 pad=SAME |
| **Transposed Convolution + Sigmoid** filters=1 size=4 stride=2 pad=SAME |

Table 1: $\beta$-VAE architecture used during experiments with the dSprites dataset.

## C QUALITATIVE AND QUANTITATIVE EXPERIMENTS WITH OUR RECONSTRUCTION ERROR

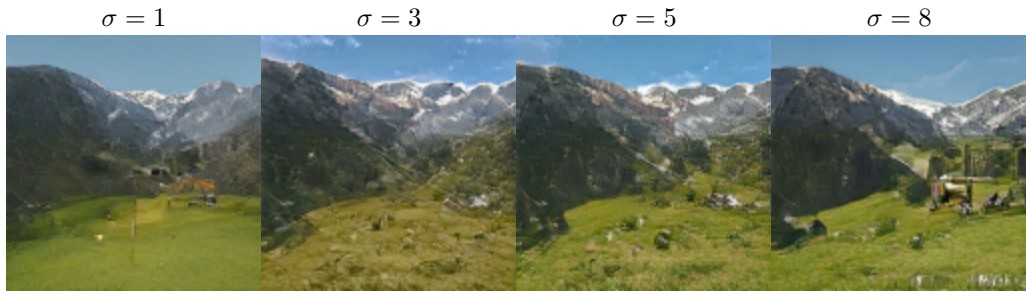

Figure 6: Reconstruction results with different $\sigma$ values. We typically used a standard deviation of 3 pixels for the kernel.

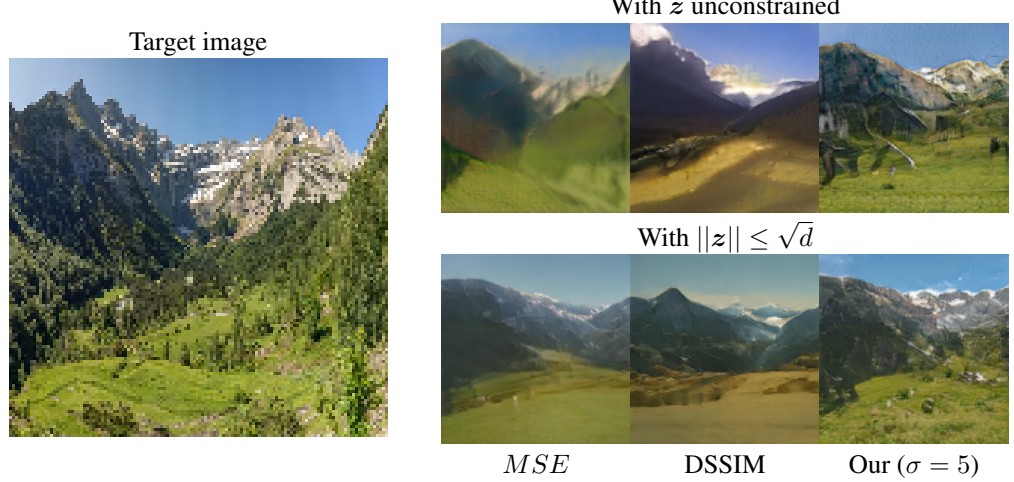

Figure 7: Reconstruction results obtained with different reconstruction errors: MSE, DSSIM (Zhou Wang et al., 2004) and our loss. With or without the constraint on $||\boldsymbol{z}||$. Note the artifacts when using our loss without constraining $\boldsymbol{z}$ (best seen with zoom).

On Fig. 6 we show qualitative reconstruction results with our method (Eq. 3) for several values of $\sigma$. On this representative example, we observe quite good results with $\sigma = 3$ and $\sigma = 5$. Higher values penalizes too low frequencies that lead to a less accurate reconstruction.

We also illustrate on Fig. 7 a comparison of our approach to two others, namely classical Mean Square Error (MSE) and Structural dissimilarity (DSSIM) proposed by Zhou Wang et al. (2004). Results are also presented with an unconstrained latent code during optimization (Eq. 1) and the approach proposed (Eq. 2). This example show the accuracy of the reconstruction obtained with our approach, as well as the fact that the restriction of $z$ to a ball of radius $\sqrt{d}$ avoids the presence of artifacts.

We also performed a quantitative evaluation of the performance of our approach. We randomly selected one image for each of the 1000 categories of the ILSVRC dataset and reconstructed it with our method with a budget of 3000 iterations. We then computed the Learned Perceptual Image Patch Similarity (LPIPS), proposed by Zhang et al. (2018), between the final reconstruction and the target image. We used the official implementation of the LPIPS paper with default parameters. Results are reported in Table 2. It suggests that images reconstructed using our reconstruction error are perceptually closer to the target image than those obtained with MSE or DSSIM. The higher standard deviation for the MSE reconstructed image LPIPS suggests that some images are downgraded in

terms of perception. It can be the case for the textured ones in particular, for the reasons explained in the Section A.

| reconstruction error | mean LPIPS | std LPIPS |
|---|---|---|
| MSE | 0.57 | 0.14 |
| DSSIM | 0.58 | 0.12 |
| **Our** ($\sigma = 3$) | **0.52** | **0.12** |

Table 2: Perceptual similarity measurements between an image and its reconstruction for different reconstruction errors.

## D  ON THE DIFFICULTY OF OPTIMISATION ON THE NATURAL IMAGE MANIFOLD.

The curvature of the natural image manifold makes the optimisation problem of Equation 2 difficult to solve. This is especially true for factors of variation which correspond to curved walks in pixel-space (for example translation or rotation by opposition to brightness or contrast changes which are linear).

To illustrate this fact, we show that the trajectory described by an image undergoing common transformations is curved in pixel space. We consider three types of transformations, namely translation, rotation and scaling, and get images from the dSprites (Matthey et al., 2017) dataset which correspond to the progressive transformation (interpolation) of an image. To visualize, we compute the PCA of the resulting trajectories and plot the trajectories on the two main axes of the PCA. The result of this experiment can be seen in Figure 8.

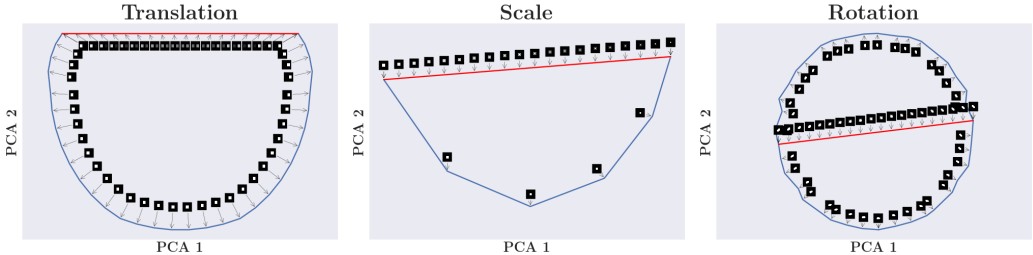

Figure 8: Two trajectories are shown in the pixel space, between an image and its transformed version, for three types of transformations: translation, scale and orientation. Red: shortest path (interpolation) between the two extremes of the trajectory. Blue: trajectory of the actual transformation. At each position along the trajectories, we report the corresponding image (best seen with zoom).

In this figure, we can see that for large translations, the direction of the shortest path between two images in pixel-space is near orthogonal to the manifold. The same problem occurs for rotation and, at a smaller extent, for scale. However this problem does not exist for brightness for example, as its change is a linear transformation in pixel-space. This is problematic during optimization of the latent code because the gradient of the reconstruction loss with respect to the generated image is tangent to this direction. Thus, when we are in the case of near orthogonality, the gradient of the error with respect to the latent code is small.

Indeed, let us consider an ideal case where $G$ is a bijection between $\mathcal{Z}$ and the manifold of natural images. Let be $\boldsymbol{z} \in \mathcal{Z}$, a basis of vectors tangent to the manifold at point $G(\boldsymbol{z})$ is given by $\left( \frac{\partial G(\boldsymbol{z})}{\partial \boldsymbol{z}_1}, ..., \frac{\partial G(\boldsymbol{z})}{\partial \boldsymbol{z}_d} \right)$.

If $\nabla_{G(\boldsymbol{z})} \mathcal{L}(G(\boldsymbol{z}), I_{\text{target}})$ is near orthogonal to the manifold then:

$$\forall i \in 1, ..., d : \langle \nabla_{G(\boldsymbol{z})} \mathcal{L}(G(\boldsymbol{z}), I_{\text{target}}), \frac{\partial G(\boldsymbol{z})}{\partial \boldsymbol{z}_i} \rangle = \epsilon_i \text{ with } \epsilon_i \approx 0 \tag{14}$$

Thus,

$$\|\nabla_z \mathcal{L}(G(z), I_{\text{target}})\| = \left\|\frac{\partial G(z)}{\partial z}^* \nabla_{G(z)} \mathcal{L}(G(z), I_{\text{target}})\right\| = \sqrt{\sum_{i=1}^{d} \epsilon_i^2} \approx 0 \qquad (15)$$

It shows that when the direction of descent in pixel space is near orthogonal to the manifold described by the generative model, optimization gets slowed down and can stop if the gradient of the loss with respect to the generated image is orthogonal to the manifold.

For example, let assume we have an ideal GAN which generates a small white circle on a black background, with a latent space of dimension 2 that encodes the position of the circle. Let consider a generated image with the circle on the left of the image and we want to move it to the right. Obviously, we thus have $\nabla_z \|G(z) - \mathcal{T}_T(G(z_1))\|^2 = 0$ if the intersection of the two circles is empty (see Figure 8) since a small translation of the object does not change the reconstruction error.

# E  ADDITIONAL QUALITATIVE EXAMPLES

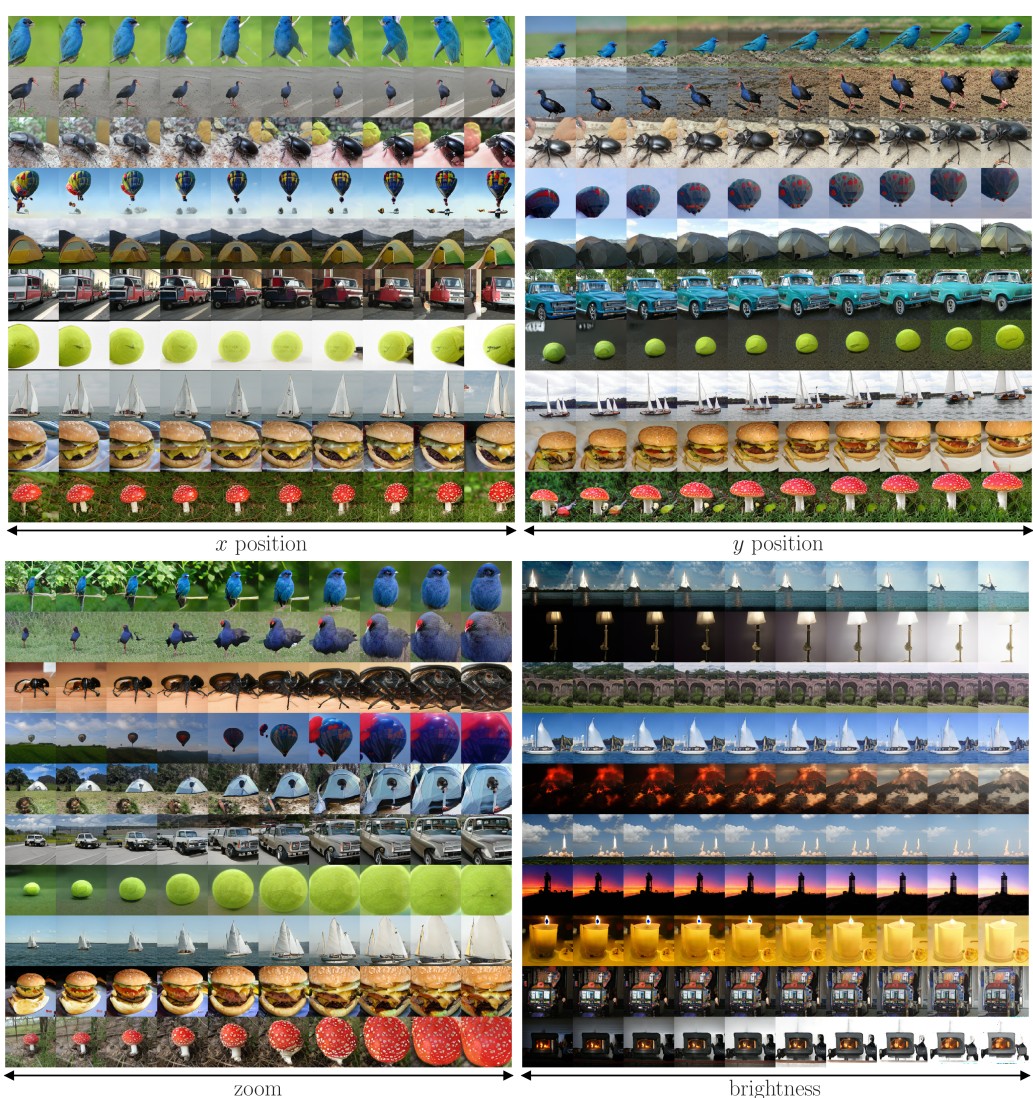

Figure 9: Qualitative results for 10 categories of ILSVRC dataset for three geometric transformations (horizontal and vertical translations and scaling) and for brightness.

We show qualitative examples for images generated with the BigGAN model for position, scale and brightness. The images latent codes are sampled in the following way: $z - \langle z, u \rangle u + \alpha u$ with $\alpha \in [-3, 3]$ and $u$ the learned direction. We have chosen the categories to produce interesting results: for position and scale categories are objects, for brightness categories are likely to be seen in a bright or dark environment. Notice that for some of the chosen categories, we failed to control the brightness of the image. It is likely due to the absence of dark images for these categories in the training data. for position and scale, the direction is learned on the ten categories presented here while for brightness only the five top categories are used.

## F  QUALITATIVE COMPARISON BETWEEN OUR OPTIMIZATION METHOD AND THE NAIVE METHOD.

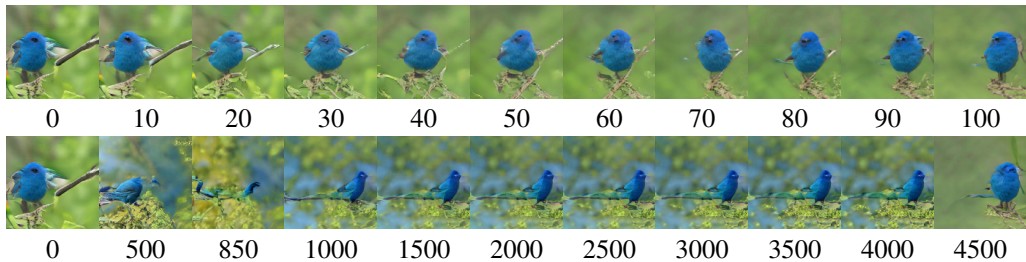

Figure 10: Comparison of the speed of convergence on a single example for our method (top) given by equation 5 and a naive approach (bottom) given by equation 4. The numbers indicate the step of optimization. Both experiences have been conducted with Adam optimizer with a learning rate of $1e{-}1$.

