# OpenReview forum: "Controlling generative models with continuous factors of variations"
_ICLR.cc/2020/Conference — Accept (Poster)_

### Official Review · AnonReviewer3 · 2019-10-20
**Official Blind Review #3**

**Rating:** 6

**Review:**

This paper proposes a method to learn and control continuous factors of variations within generative models by finding meaningful directions in the latent space which correspond to specified properties. A new method is proposed for inverting generative models and embedding images in the latent space when an encoder is not available. Specifically, reconstruction error is defined in the Fourier domain such that the weighting on high frequency image components can be reduced. Results are evaluated with qualitative comparison to previous embedding methods. Using this image embedding technique, a dataset of latent space trajectories is created by manipulating a desired property in images (such as position or scale) via affine transformations and recording the latent space vectors of the original and new images. The dataset is then used to learn a simple model of the latent space transformation corresponding to changes in the desired image property, which in turn can be used to manipulate images accordingly. To evaluate the effectiveness of this image manipulation approach, a saliency detector is used to measure the change in position or scale of objects in generated images as the latent codes are changed.

Overall, I would tend towards accepting this work. The goal of being able to manipulate continuous factors of variation within generative models is useful for controllable image synthesis, and the proposed method clearly achieves the desired result.


Things to improve the paper:
1) The paper proposes a new reconstruction error metric which is optimized to embed images into the latent space of the generative models. While this new metric is compared qualitatively to existing methods, quantitative evaluation is lacking. It would be useful to also include quantitative comparison of methods measuring the perceptual distance between the original image and the embedded image, perhaps by using Learned Perceptual Image Patch Similarity (LPIPS) [1].


Minor things to improve the paper that did not impact the score:
2) In the abstract: "Our method is weakly supervised...". I am not sure if this method would be considered weakly supervised. I might tend more towards calling it self-supervised, since we have exact labels that are derived from transformations applied to the images themselves.

3) In the first paragraph of the introduction: "an increasing number of applications are emerging such as image in-painting, dataset-synthesis, deep-fakes... ". I find the use of the ellipses here to be a bit strange, since it seems like the sentence is trailing off mid-thought. I would recommend the use of "etc." over "...".

4) In Section 2.2, second paragraph, the dSprite dataset is mentioned but not cited. The reference is not given until Section 3. Should the citation be paired with the first mention of the dataset? Or even just in both places.

5) In Section 3, Implementation details: "The first part is injected at the bottom layer while next parts are used to modify the style of the generated image thanks to AdaIN layers (Huang & Belongie, 2017)". BigGAN uses conditional BatchNorm instead of AdaIN, although they are both very similar. I think the proper citation here is [2], which first introduced conditional BatchNorm.


Questions:
6) I am not fully convinced of the argument that using a saliency detector makes the method more general purpose than a dedicated object detector. The majority of high quality generative models are class conditional, hence requiring a labelled dataset, and therefore an object detector can easily be trained on the same dataset. Additionally, Section 3.2 mentions that "We performed quantitative analysis on ten chosen categories for which the object can be easily segmented by using saliency detection approach", which seems to indicate that the saliency detector struggles with some objects. How does the saliency detector perform on more complicated objects?


References:
[1] Zhang, Richard, et al. "The unreasonable effectiveness of deep features as a perceptual metric." Proceedings of the IEEE Conference on Computer Vision and Pattern Recognition. 2018.

[2] De Vries, Harm, Florian Strub, Jérémie Mary, Hugo Larochelle, Olivier Pietquin, and Aaron C. Courville. "Modulating early visual processing by language." In Advances in Neural Information Processing Systems, pp. 6594-6604. 2017.


### Post-Rebuttal Comments ###
Thanks you for addressing my concerns and for adding the quantitative reconstructions measures. Appendix C looks much more complete now. My overall opinion of the paper remains about the same, so I will leave my score unchanged.


**Experience Assessment:**

I have published one or two papers in this area.

**Review Assessment: Checking Correctness Of Derivations And Theory:**

I did not assess the derivations or theory.

**Review Assessment: Checking Correctness Of Experiments:**

I assessed the sensibility of the experiments.

**Review Assessment: Thoroughness In Paper Reading:**

I read the paper at least twice and used my best judgement in assessing the paper.

---

> ### Author Response · Authors · 2019-11-08
> **Relpy to reviewer 3**
>
> Thank you for your time and expertise in your review, we've addressed the key points below:
>
> > The paper proposes a new reconstruction error metric that is optimized to embed images into the latent space of the generative models. While this new metric is compared qualitatively to existing methods, quantitative evaluation is lacking. It would be useful to also include a quantitative comparison of methods measuring the perceptual distance between the original image and the embedded image, perhaps by using Learned Perceptual Image Patch Similarity (LPIPS)
>
>     We added this quantitative comparison in Appendix C using the LPIPS. We agree that it strengthens the article and mention it in Section. Current results are reported with 40 images (for fast feedback) but we plan to compute it on $1,000$ images and update the PDF in a couple of days. We used unbiased mean and standard deviation estimators. (Edit: We updated the results for 1000 images in the latest revision.)
>
> > I am not fully convinced of the argument that using a saliency detector makes the method more general-purpose than a dedicated object detector. The majority of high-quality generative models are class conditional, hence requiring a labeled dataset, and therefore an object detector can easily be trained on the same dataset.
>
>     We understand the argument, but a dedicated object detector requires labeled bounding-boxes coordinates while class conditional generative models only need a categorical label. In any case, our approach remains more generic and less computationally demanding. It is nevertheless worth noting that saliency detection is only useful for the quantitative evaluation of the method: it does not change any "level of generality" of the method itself.
>
> > Additionally, Section 3.2 mentions that "We performed quantitative analysis on ten chosen categories for which the object can be easily segmented by using saliency detection approach", which seems to indicate that the saliency detector struggles with some objects. How does the saliency detector perform on more complicated objects?
>
>     In fact, some categories of ILSVRC are not actual "objects". It is, for example, the case for "beach" or "cliff". Hence, we preferred to choose categories that are actual objects, such as dog, flower or ball. As a consequence, we expected saliency detection to work on it, and it was indeed the case. We selected categories for their "objectness" and then we used the saliency detection but we did not select categories knowing the performance of the saliency detection which turned out to be robust for the categories we experimented on.
>
>     We took your remark into consideration and replaced the sentence to be more explicit: "We performed quantitative analysis on ten chosen categories of objects of ILSVRC, avoiding non-actual objects such as "beach" or "cliff".
>
>     We also thank you for your other "Minor things to improve the paper", that we took into consideration for the updated version of the article.

---

### Official Review · AnonReviewer1 · 2019-10-22
**Official Blind Review #1**

**Rating:** 8

**Review:**

Summary:

This paper proposes methods to find interpretable vectors in the latent space of generative models (similar to finding Smile Vectors [White, 2016]) which control simple object transformations like zoom or translation. The basic idea is that so long as one can apply the desired transformation to an image, one can solve for the latent which minimizes the reconstruction between G(z) and the transformed image; doing this for various parameters of the transformation (i.e. different levels of zoom, varying amounts of brightening or translation) allows one to learn a parametric mapping specifying how to vary the latent to achieve the desired output change. The authors make several changes to the naïve optimization procedure of vanilla SGD, most notably using reconstruction error on Gaussian-blurred images to encourage matching of low-frequency features rather than high-frequency features. The resulting framework is applied to an ImageNet GAN for a variety of transformation, producing results which qualitatively and quantitatively indicate that the method works for the shown transformations, along with some analysis of the behavior of the model.

My take:

This is a well-reasoned and well-presented paper following in the spirit of Smile Vector type investigations, with compelling results. The core idea is simple, and I like that it doesn’t require human labeling: one merely needs to be able to simulate some approximation of the desired transform, and one can find the latent space trajectory that corresponds to the model’s approximation of that transform. I think this is promising  next step in this area (there have been a few papers very recently on it, so I think improving constraints and control of generative models is getting a decent amount of intention) and is worthy of acceptance at ICLR2020 (7/10; reasonably clear accept).

Notes:

-“Sampling Generative Models” (White, 2016, https://arxiv.org/abs/1609.04468) should be cited and discussed, and ideally so should “Latent constraints: Learning to generate conditionally from unconditional generative models” (Engel et al, 2017, https://arxiv.org/abs/1711.05772)--both are quite relevant IMO.

Minor:

The first sentence ends with an ellipsis. Is this intentional or a draft holdover? Either way I think it should at least be replaced with an ‘etc’ or ideally an oxford comma and an ‘and’.



**Experience Assessment:**

I have published in this field for several years.

**Review Assessment: Checking Correctness Of Derivations And Theory:**

I carefully checked the derivations and theory.

**Review Assessment: Checking Correctness Of Experiments:**

I carefully checked the experiments.

**Review Assessment: Thoroughness In Paper Reading:**

I read the paper thoroughly.

---

> ### Author Response · Authors · 2019-11-08
> **Reply to reviewer 1**
>
> Thank you for your review, we have addressed your remarks below:
>
> > -“Sampling Generative Models” (White, 2016, https://arxiv.org/abs/1609.04468) should be cited and discussed, and ideally so should “Latent constraints: Learning to generate conditionally from unconditional generative models” (Engel et al, 2017, https://arxiv.org/abs/1711.05772)--both are quite relevant IMO.
>
>     We thank you for these references that are indeed relevant. They were added to the updated manuscript. Both are presented and discussed in the related work.
>
> Minor:
>
> > The first sentence ends with an ellipsis. Is this intentional or a draft holdover? Either way, I think it should at least be replaced with an ‘etc’ or ideally an oxford comma and an ‘and’.
>
>     It has been changed in the updated version of the article.

---

### Official Review · AnonReviewer2 · 2019-10-23
**Official Blind Review #2**

**Rating:** 6

**Review:**

The paper proposes an algorithm to find linear trajectories in the latent space of a generative model that correspond to a user-specified transformation T in image space. Roughly, the latent trajectory is obtained by inverting the generator at the transformed image and a clever recursive estimation strategy is proposed to overcome difficulties in this nonconvex optimization. Qualitative results of the method, applied to a (pretrained) BigGAN model are shown, where the transformations are chosen as translation, zoom or brightness. A quantitative evaluation is performed on the dSprites and ILSVRC dataset.

My take:
There seem to be a lot of errors and typos in the manuscript, which made the paper unfortunately a bit frustrating to review. In particular, I had trouble following and understanding the details of the main procedure used to obtain the linear latent trajectories. Considering the recent works (Goetschalckx et al, 2019) and (Jahanian et al, 2019), I also don't see too much novelty in this approach. Therefore, I cannot recommend acceptance of this paper at this point.

Details:

1) In algorithm 1, there seem to be some typos which makes it difficult understand the method in detail. z_{\delta t} is initialized as z_0 and then never changed but always appended into the data set. Should it maybe be z_{\delta t} <- argmin ... instead of z_t <- argmin ... ? But then why initialize z_{\delta t} at all? Why store tuples of three values in D, especially store z_0 multiple times?

While it is clear, formally the method always discards D and one might add a D_i <- D at the end.

2) I could not follow the reasoning in Section 2.2 and the clarity should be improved, as it is one of the main contributions of the work. In particular, I would like to see the intuition behind the model t = g(<u, z>) better described.

Why does the projection of z follow a normal distribution? Is it because the latent distribution in the GAN is chosen as a normal distribution?

What is the loss for training f_{\theta,u}? How is the dataset D used here?

Minor comments / typos / suggestions (no influence on my rating):
- InfoGAN (Chen et al., 2016) does not require a labeled dataset, the corresponding sentence in related work should be reformulated a bit.
- Please use \operatorname or \text in math mode for operators such as Var or text.
- 'Encodes a the parameter t' --> 'Encodes the parameter t'; Many other typos, please run a spell checker.
- For image translation, what boundary conditions are used? A sensible way would be to impose the reconstruction loss not on the full image but only on the smaller part.


**Experience Assessment:**

I have published one or two papers in this area.

**Review Assessment: Checking Correctness Of Derivations And Theory:**

I assessed the sensibility of the derivations and theory.

**Review Assessment: Checking Correctness Of Experiments:**

I assessed the sensibility of the experiments.

**Review Assessment: Thoroughness In Paper Reading:**

I read the paper at least twice and used my best judgement in assessing the paper.

---

> ### Author Response · Authors · 2019-11-08
> **Reply to reviewer 2 on concerns raised in 1)**
>
> We thank you for the fruitful comments and suggestions. In addition to the lightly revised manuscript, we respond directly to the comments below.
>
> > There seem to be a lot of errors and typos in the manuscript, which made the paper unfortunately a bit frustrating to review. In particular, I had trouble following and understanding the details of the main procedure used to obtain the linear latent trajectories.
>
>     We indeed fixed a couple of typos thanks to the three reviewers' feedback. We answer to those you highlighted below the detailed comments you provided.
>
> > Considering the recent works (Goetschalckx et al, 2019) and (Jahanian et al, 2019), I also don't see too much novelty in this approach. Therefore, I cannot recommend acceptance of this paper at this point.
>
>     These two works have been released on arXiv as non-peered-reviewed report. Hence, we thought that they could not be considered as actual articles that are part of the scientific literature yet at the time of the ICLR 2020 deadline. This is implicit in the "dual submission policy" of ICLR and explicit in the reviewer guidelines of conferences such as CVPR. Actually, (Goetschalckx et al, 2019) has been published at ICCV last week and (Jahanian et al, 2019) has been submitted to ICLR 2020 as well (available on openreview).
>
>     Since we have heard of these two arXiv reports a couple of weeks before the ICLR deadline, we obviously mentioned them and compared our work to the idea of (Jahanian et al, 2019) that is the closest to our work. It is discussed in the related works (Section 4) and some differences are highlighted. It appears that although both works have been developed independently and concurrently, they exhibit some similarities. But since both are submitted to ICLR 2020, we think it rather enforces that the general idea is novel and relevant.
>
> > In algorithm 1, there seem to be some typos which makes it difficult understand the method in detail. $z_{\delta t}$ is initialized as $z_0$ and then never changed but always appended into the data set. Should it maybe be $z_{\delta t} <- argmin $... instead of $z_t <- argmin ...$ ?
>
>     Indeed, the $\delta$ is missing: it is not $z_t$ but $z_{\delta t}$. It has been changed in the updated version of the article.
>
> > But then why initialize $z_{\delta t}$ at all?
>
>     Concerning the initialization of $z_{\delta t}$ we initialize it at $z_0$ as $z_0$ is expected to be close to the solution of the first optimization problem: $argmin_{z}\mathcal{L}(G(z), \mathcal{T}_{\delta t_1}(I_0))$. It is thus the initialization of the recursive procedure presented in Section 2.1.2 (and Equation 5).
>
> > Why store tuples of three values in $D$,
>
>         During the manuscript redaction, we hesitated on that point. Indeed, it would not have been necessary to store the three values to estimate the trajectory only. However later, our method uses all these three values to train the model in Section 2.2. Thus, we chose to present Algorithm 1 as a method to create the full required dataset. We nevertheless admit that one can use the method of Section 2.1.2 to estimate a trajectory only, and thus retain only $z_{\delta t}$ in D only. We added a mention to this in the caption of Algorithm 1.
>
> > especially store $z_0$ multiple times?
>
>     $z_0$ is different for each trajectory and we need to store it along $z_{\delta t}$ and $\delta t$ to be able to train our model later.
>
> > While it is clear, formally the method always discards D and one might add a $D_i \leftarrow D$ at the end.
>
>     Indeed there should be a $D_i$ for each trajectory. Since we only need a dataset of trajectories, we propose to initialize $D$ before the for loop. It has been changed in the updated version of the article.

---

> ### Author Response · Authors · 2019-11-08
> **Reply to reviewer 2 on concerns raised in 2) and on the minor comments**
>
>
> > I could not follow the reasoning in Section 2.2 and the clarity should be improved, as it is one of the main contributions of the work. In particular, I would like to see the intuition behind the model $t = g(<u, z>)$ better described.
>
>     Thank you for pointing this out we reformulated the explanation in the updated manuscript. The synthetic explanation is the following:
>
>     A core hypothesis is that we can modify $t$ by moving along a direction $u$ in the latent space thus the model that predict $t$ from $z$ should be a function of $<z, u>$. However, despite the popularity of a model of the form $t = <z, u>$, it is only adapted if $t$ follows a normal distribution (in the common case where $z$ is sampled from a Gaussian distribution). Indeed, if $z$ follows a normal distribution, the prediction of $t$ will also follow such distribution. It is thus problematic if $t$ does not follow this type of distribution for the images actually generated by the model. Thus we propose to use a more general (parametrized) model of the form  $t = g_{\theta}(<z, u>)$. It  is coherent with the initial hypothesis while allowing to have a good fit even when $t$ does not follow a normal distribution.
>
> > Why does the projection of z follow a normal distribution? Is it because the latent distribution in the GAN is chosen as a normal distribution?
>
>     Yes, the latent distribution in the GAN usually follows a normal distribution in the literature, thus its projection on a linear space follow a Gaussian too. We reformulated this in the updated manuscript. Combined with the change due to your preceding remark, it indeed clarifies the explanation.
>
> > What is the loss for training $f_{\theta,u}$ ? How is the dataset $D$ used here?
>
>     It is a regression problem, we used the MSE and trained it from the dataset with the tuples $(z_0, z_{\delta t}, \delta t)$. It has been mentioned in the updated version of the article.
>
> Minor comments / typos / suggestions (no influence on my rating):
>
> > InfoGAN (Chen et al., 2016) does not require a labeled dataset, the corresponding sentence in related work should be reformulated a bit.
>
>     Indeed, it has been changed in the updated version of the article. We also extended the discussion w.r.t to it to clearly differentiate our work.
>
> > Please use operatorname or text in math mode for operators such as Var or text.
>
>     It has been changed in the updated version of the article.
>
> > 'Encodes a the parameter $t$ $-->$ 'Encodes the parameter $t$; Many other typos, please run a spell checker.
>
>     We proofread the manuscript.
>
> > For image translation, what boundary conditions are used? A sensible way would be to impose the reconstruction loss not on the full image but only on the smaller part.
>
>     In Section 2.1.2 we mentioned:
>     "A transformation on an image usually leads to undefined regions in the new image (for instance, for a translation to the right, the left hand side is undefined). This is why we designed $\mathcal{L}$ to ignore the value of the undefined regions of the image"
>
>     We simplified the last sentence to be more explicit: "Hence, we ignore the value of the undefined regions of the image to compute  $\mathcal{L}$."

---

### Decision · Program_Chairs · 2019-12-19

**Decision:**

Accept (Poster)

**Comment:**

Following the revision and the discussion, all three reviewers agree that the paper provides an interesting contribution to the area of generative image modeling. Accept.